# Identification of Putative Neuropeptides That Alter the Behaviour of *Schistosoma mansoni* Cercariae

**DOI:** 10.3390/biology11091344

**Published:** 2022-09-12

**Authors:** Conor E. Fogarty, Saowaros Suwansa-ard, Phong Phan, Donald P. McManus, Mary G. Duke, Russell C. Wyeth, Scott F. Cummins, Tianfang Wang

**Affiliations:** 1Centre for Bioinnovation, University of the Sunshine Coast, Maroochydore, QLD 4556, Australia; 2School of Science, Technology and Engineering, University of the Sunshine Coast, Maroochydore, QLD 4556, Australia; 3QIMR Berghofer Medical Research Institute, Brisbane, QLD 4006, Australia; 4Department of Biology, St. Francis Xavier University, Antigonish, NS B2G 2W5, Canada

**Keywords:** schistosomiasis, *Schistosoma mansoni*, cercariae, putative neuropeptide, behavioural bioassay, parasite–host interaction

## Abstract

**Simple Summary:**

*Schistosoma mansoni* is a common etiological agent of human schistosomiasis, one of the deadliest neglected tropical diseases in the developing world. *Schistosoma* neuropeptides have been considered a possible avenue of targeted biocontrol. We aimed to identify species-specific putative neuropeptides derived from the mammalian-infective stage, cercariae, which may function as biocontrol targets. A total of 11 neuropeptide precursors were identified for the first time in cercariae, all of which were highly specific to *Schistosoma* and many of which were highly expressed in the cercarial stage. We tested nine putative neuropeptides derived from these precursor proteins, three of which induced a significant increase in cercariae stopping, turning and passive behaviour over periods of both 1 min and 360 min post-exposure. These observations implicate the tested neuropeptides as involved in regulating cercarial behaviour. The characterisation of these putative neuropeptides may facilitate the innovation of biocontrols to prevent infection.

**Abstract:**

Elucidating the infectivity of *Schistosoma mansoni*, one of the main etiological agents of human schistosomiasis, requires an improved understanding of the behavioural mechanisms of cercariae, the non-feeding mammalian infective stage. This study investigated the presence and effect of cercariae-derived putative neuropeptides on cercarial behaviour when applied externally. Cercariae were peptidomically analysed and 11 neuropeptide precursor proteins, all of which were specific to the *Schistosoma* genus and most of which highly expressed in the cercarial stage, were identified in cercariae for the first time. Protein–protein interaction analysis predicted the interaction of various neuropeptide precursors (e.g., Sm-npp-30, Sm-npp-33, Sm-npp-35) with cercarial structural proteins (e.g., myosin heavy chain and titin). In total, nine putative neuropeptides, selected based on their high hydrophobicity and small size (~1 kilodalton), were tested on cercariae (3 mg/mL) in acute exposure (1 min) and prolonged exposure (360 min) behavioural bioassays. The peptides AAYMDLPW-NH_2_, NRKIDQSFYSYY-NH_2_, FLLALPSP-OH, and NYLWDTRL-NH_2_ stimulated acute increases in cercarial spinning, stopping, and directional change during active states. However, only NRKIDQSFYSYY-NH_2_ caused the same behavioural changes at a lower concentration (0.1 mg/mL). After prolonged exposure, AAYMDLPW-NH_2_ and NYLWDTRL-NH_2_ caused increasing passive behaviour and NRKIDQSFYSYY-NH_2_ caused increasing body-first and head-pulling movements. These findings characterise behaviour-altering novel putative neuropeptides, which may inform future biocontrol innovations to prevent human schistosomiasis.

## 1. Introduction

Schistosomiasis is a disease caused by infection with helminthic parasites of the genus *Schistosoma* and is one of the most substantial threats to public health in the developing world. *Schistosoma* are digenetic trematodes which alternate between intermediate molluscan and definitive mammalian hosts. Over 20 million people suffer from symptoms of severe schistosomiasis, including serious hormonal, physiological, and immunological complications [1,2,3,4]. Infection can reduce resistance to HIV and malaria and increase rates of anaemia, malnutrition, and delayed development in children [5,6,7]. The main method of managing schistosomiasis is praziquantel, a highly efficacious chemotherapeutic drug that produces few serious side effects [8,9]. However, it is inaccessible to over 70% of those in need and there are concerns that over-reliance may lead to the evolution of resistant parasites [10,11]. Furthermore, mass drug administration may be an insufficient long-term solution because unless the parasites are removed from the surrounding environment infection rates tend to rebound rapidly [12]. The risks and shortcomings of an over-reliance on chemotherapy necessitate the investigation of other schistosomiasis-inhibiting methods.

One alternate approach worth consideration is interference with schistosome cercariae, the mammalian-infective stage. Trematode cercariae movement towards their host relies on a mixture of chemical, thermal, and luminous gradients, with variations depending on the species [13,14]. Cercariae from *Schistosoma mansoni*, one of the most widely studied human-infective schistosome species, have been shown to display chemotactic behaviour towards mammalian host proteins [15]. This response may be mediated by G protein-coupled receptors (GPCRs). To date, 36 GPCRs have been identified with the help of the *S. mansoni* genome [16,17], some of which bind neuropeptides.

Neuropeptides are among the most structurally and functionally diverse classes of metazoan signalling molecules, playing key roles in the regulation of many vital physiological processes and behaviours [18]. Neuropeptides are highly conserved across the *S. mansoni* lifecycle, involved in essential processes such as locomotion and metabolism, and are potentially directly involved in host recognition [19]. A recent study analysed neuropeptides in miracidia (the molluscan-host infective stage); these are possibly involved in host-identification responses and could function as targets for biocontrols to prevent molluscan infection [20]. Similarly, neuropeptides could function as potential targets for interfering with cercariae infectivity and survival [21,22]. An avenue of neurobiology interference—antagonism—was demonstrated as highly efficacious by a molecule called “Schistosome Paralysis Factor”. This chemical, suspected to antagonise serotonin signalling, significantly decreased cercariae survival and infectivity at concentrations as low as 25 nM [23]. However, there remain substantial gaps in our knowledge of neuropeptides in schistosomes, making it difficult to accurately determine their functions and hence assess their viability as targets for biocontrols [24].

This study focused on identifying and analysing the role of putative neuropeptides in regulating the behaviour of *S. mansoni* cercariae. This involved performing a peptidomic and in silico analysis to identify putative neuropeptides. Subsequently, behaviour bioassays were employed to assess the bioactivity of selected putative neuropeptides on cercarial movement. The identification of novel putative neuropeptides and their effects on cercarial behaviour explored in this study further elucidate cercarial neurobiology and may facilitate the development of novel control strategies.

## 2. Materials and Methods

### 2.1. Biomphalaria glabrata Maintenance Conditions and Ethics Guidelines

The life cycle of *Schistosoma mansoni* was maintained at QIMR Berghofer Medical Research Institute (QIMRB), Brisbane, under an Australian Department of Agriculture, Fisheries and Forestry Biosecurity permit. All work was carried out under strict animal ethics requirements. *Biomphalaria glabrata* snails of the NMRI strain were maintained in an aerated tank of calcium carbonate conditioned water (pH-neutral) at 27 °C in a 12 h alternating cycle of light and darkness. The snail diet consisted of algae tablets and lettuce. The conduct and procedures involving animal experimentation were approved by the Animal Ethics Committee of the QIMRB (project number P3705). This study was performed in accordance with the recommendations in the Guide for the Care and Use of Laboratory Animals of the National Institutes of Health.

### 2.2. Schistosoma mansoni Cercaria Collection and Peptide Isolation

NMRI strain *B. glabrata* snails, infected 4 weeks prior, were incubated in 3 beakers containing 20 mL of pH-neutral water at room temperature for cercarial shedding. The water was collected after 2 h and contained an estimated 6000 cercariae. The water samples were centrifuged at 4000× *g* at 4 °C for 15 min. The supernatant was removed and the pellet of cercariae was homogenised thoroughly in a 1:4 *v/v* ratio of acidified methanol (Sigma-Aldrich, St. Louis, MO, USA; methanol/glacial acetic acid/MilliQ water, 90:9:1). The mixture was centrifuged at 12,000× *g* at 4 °C for 15 min. The supernatant was extracted and lyophilised using a Savant SpeedVac Concentrator (Thermo Scientific, Waltham, MA, USA).

### 2.3. uHPLC Tandem QTOF MS/MS Analyses

Lyophilised samples were resuspended in 35 µL of 0.1% (*v/v*) formic acid in MilliQ water and analysed by LC-MS/MS with an ExionLC liquid chromatography system (AB SCIEX, Concord, ON, Canada) and a QTOF X500R mass spectrometer (AB SCIEX, Concord, ON, Canada) equipped with an electrospray ion source, as described elsewhere [25]. A 25 µL sample was injected into a 100 mm × 1.7 μm Aeris PEPTIDE XB-C18 100 uHPLC column (Phenomenex, Sydney, Australia) equipped with a SecurityGuard column. Linear gradients of 3–35% solvent B over a 10-min period at a flow rate of 400 µL/min, followed by a gradient from 35% to 80% solvent B over 2 min and 80% to 95% solvent B over 1 min were used for peptide elution. Solvent A consisted of 0.1% formic acid in MilliQ water while solvent B contained 0.1% formic acid in 100% acetonitrile. The ion spray voltage was set to 5500 V, the declustering potential was set to 100 V, the curtain gas flow was set at 30, the ion source gas 1 was set at 40, the ion source gas 2 was set at 50, and the spray temperature was set at 450 °C. The mass spectrometer acquired the mass spectral data in an Information Dependant Acquisition (IDA) mode. Full-scan TOFMS data were acquired over the mass range of 350–1400 and, for product ion, ms/ms 50–1800. Ions observed in the TOF-MS scan exceeding a threshold of 100 cps and a charge state of +2 to +5 were set to trigger the acquisition of product ion. The data were acquired and processed using SCIEX OS software (AB SCIEX, Concord, ON, Canada).

### 2.4. Protein Identification

LC-MS/MS raw data were converted by the MSConvert module of ProteoWizard (3.0.1) [26] and imported to PEAKS studio (Bioinformatics Solutions Inc., Waterloo, ON, Canada, version 7.0). The *S. mansoni* protein database downloaded from Wormbase (https://parasite.wormbase.org/Schistosoma_mansoni_prjea36577/Info/Index, accessed on 1 August 2020) was used in the data analysis. De novo sequencing of peptides, database searches, and characterising specific post-translational modifications (PTMs) were used to analyse the raw data; false discovery rate (FDR) was set to ≤1%, and [−10 × log(*P*)] was calculated accordingly where *P* was the probability that an observed match was a random event. The PEAKS used the following parameters: (i) precursor ion mass tolerance, 20 ppm; (ii) fragment ion mass tolerance, 0.1 Da (the error tolerance); (iii) no enzyme was selected; (iv) monoisotopic precursor mass and fragment ion mass; and (v) variable modifications including N-terminal acetylation, deamidation on asparagine and glutamine, oxidation of methionine, conversion of glutamic acid and glutamine to pyroglutamate, and C-terminal amidation.

### 2.5. Prediction of Putative Neuropeptides, Gene Ontology, and KEGG Pathway Analysis

Identified proteins were analysed using BLASTp of the National Center for Biotechnology Information (NCBI) to search against the non-redundant protein database. Protein N-terminal signal sequences were predicted using SignalP 4.1 [27] and PrediSi [28], with the transmembrane (TM) domains predicted by TMHMM [29]. For SignalP predictions, positive identifications were made when both neural network and hidden Markov model algorithms gave coincident estimations; D-cutoff values were set to 0.34 (to increase sensitivity) for both SignalP-noTM and TM networks. BLAST results were combined and imported to BLAST2GO [30] (version 5.1) to perform gene ontology (GO) and KEGG pathway analysis. The putative neuropeptides were predicted by NeuroPred [31] and the cleavage sites of their precursors were obtained.

### 2.6. Protein–Protein Interaction (PPI) Network

We investigated the PPI maps following a similar procedure reported elsewhere [32]. The PPI between annotated *S. mansoni* proteins identified and the entire proteome was revealed by STRING [33]. STRING integrates protein–protein interactions from multiple resources, including direct (physical) as well as indirect (functional) associations. Because physical interactions were not established in previous studies and could not be confirmed in this study, only indirect associations were used, including co-expression and predicted domain–domain interaction. All resources were selected to generate the network and ‘confidence’ was used as the meaning of network edges. The minimum combined interaction score was set to 0.15 to increase sensitivity. Here, interactions derived from co-expression were defined as when the genes showed a similar pattern of mRNA expression, which was measured by DNA arrays and similar technologies in previous reports. Proteins without any interaction with other proteins were excluded from the network. Topological and statistical analyses were performed to explore the potential functions in our constructed network using the NetworkAnalyzer plugin in Cytoscape 3.7.1 [33]. The final network was visualised using Cytoscape [34].

### 2.7. Comparative Sequence Analysis of Putative Neuropeptides

Annotation of putative *S. mansoni* neuropeptides was performed by BLASTp searching against the NCBI non-redundant protein database (21 May 2020). Homologous proteins matching with putative *S. mansoni* neuropeptides with *E*-values ≤ 0.5 were retrieved from NCBI and used for multiple sequence alignments. Amino acid alignment was generated using MEGA X software (version 10.1.8) [35] with parameters set as follows: algorithm, ClustalW; gap opening penalty, 10; gap extension penalty, 0.2. Visualisation of the alignment was carried out on TeXworks software.

### 2.8. Peptide Synthesis and Preparation for Bioassay

Neuropeptide precursor proteins extracted from *S. mansoni* cercariae and their respective peptides were considered for further study based on several criteria. The precursors were required to have high specificity to *Schistosoma* (according to the BLASTp search against the NCBI protein database) and relatively highly expression in the cercarial stage [36]. Additionally, the peptides were required to be relatively small (>2 kilodaltons) to increase the probability of penetration during incubation. Based on these criteria, 9 peptides, shown in Table 1, were selected and synthesised by ChinaPeptides Co., Ltd. (Shanghai, China) for behavioural assays. These included 5 peptides derived from novel precursor proteins identified in this study (Sm-npp-30, Sm-npp-33, Sm-npp-35, Sm-npp-36, and Sm-npp-38) and 4 peptides derived from established neuropeptide precursors (Sm-npp-5, Sm-npp-14, Sm-npp-17, and Sm-npp-26). Of the 9 peptides, 2 were predicted exclusively from NeuroPred (NP4 and NP9) and 7 were supported by both NeuroPred and MS/MS spectra (NP1, NP2, NP3, NP5, NP6, NP7, and NP8). The purity of the peptides was ≥95% as determined by reverse-phase HPLC. Prior to bioassays, peptides were dissolved in MilliQ water to a concentration of 3 mg/mL.

### 2.9. Schistosoma mansoni Cercarial Behavioural Bioassays

Cercariae were shed from *B. glabrata* and the total number of cercariae was estimated from 100 µL aliquots under a compound microscope. An estimated 10,000 cercariae in 20 mL were employed in this bioassay. Water aliquots in 100 µL volumes containing cercariae were placed on glass slides (StarFrost^®^ superclean, hydrophilic slides, ground edges 90°, white, ProSciTech Pty Ltd., Kirwan, Australia) and monitored using an Olympus-CKX41 microscope (Olympus) equipped with an Olympus DPI Digital Microscope Camera DP22 (15 frames per second, at 2.8-megapixel image quality). A 400× magnification was employed for the analysis, resulting in a field of view (FOV) of 4.60 mm × 3.45 mm (400 × 300 pixels; 1 pixel = 11.5 µm). For one min, cercarial behaviour was recorded and monitored (designated as a pre-exposure behaviour), a 2 µL aliquot of the peptide solution was added, and cercarial behaviour (designated as a post-exposure behaviour) was recorded for another min within the FOV. MilliQ water was used as a negative control. To test responses to acute exposure, all putative neuropeptides were initially tested at 3 mg/mL in replicates of 9 (final concentration of 59 µg/mL in the aliquot, although higher concentrations were present initially as the putative neuropeptide solution had diffused following addition to the aliquot). NP2 could not be tested due to its excessively high hydrophobicity. Two putative neuropeptides (NP5 and NP7) and the control had their respective sample sizes increased from 9 to 20 to increase the clarity of their effects on behavioural changes following preliminary bioassays. Putative neuropeptides that produced significant changes in behaviour relative to the control were also tested at concentrations of 0.1 mg/mL (final concentration of 2 µg/mL).

### 2.10. Behavioural Bioassay Data Analysis

#### 2.10.1. Analysis of the Characteristics of Cercarial Movement

Bioassay videos were quantitatively analysed and the effects of putative neuropeptides on cercarial behaviour were determined statistically using the method described in detail elsewhere [32]. Using a rolling mean subtraction method described elsewhere [37], the contrast between cercariae and the background was improved. Cercariae x-y locations were tracked in each frame and the tracks were interpolated using the TrackMate plugin [38] in ImageJ software [39]. In addition, tracks for all pre- and post-exposure videos were manually reviewed and adjusted as necessary in the MTrackJ plugin [40] to connect disjointed tracks, remove cercarial tail data, and eliminate tracks created by non-target objects. Cercariae displayed two distinct behavioural states: active states and passive states. Active states were characterised by the cercariae moving in a predominately tail-first direction and passive states were characterised by cercariae immobility and occasional head elongation [41]. For additional clarity, we calculated different metrics for both active and passive states. We identified points in the tracks as passive when at least 7 of the previous 10 successive points in a track were less than 17 µm distant. Cercariae were categorised as in a passive state when at least 10 consecutive points were passive points. The relative number of passive points and states per track were calculated by dividing the sum of passive points and states by the total number of points in each track, respectively. The average velocity of the heads of cercariae were calculated exclusively in active states. The angular standard deviation (SD) was calculated from the change in angles between consecutive points in active states, indicating the magnitude and frequency of angle changes. Therefore, lower angular SD corresponds to straight movement while higher angular SD indicates cercarial wandering. All behavioural measurements were compared between the pre- and post-exposure data. Replicates in which no cercariae were present were removed from calculations.

Preliminary analyses with repeated measures MANOVAs and linear mixed effects models both failed to meet assumptions for normality. Thus, we completed an aligned rank transform ANOVA (ART ANOVA), with peptide as the between-subjects factor and pre/post-exposure as the within-subjects factor [42]. This was followed by post hoc contrasts of the changes between pre- and post-exposure for all putative neuropeptide treatments against MilliQ water [43]. A change of *p <* 0.05 was considered significant. Statistical analysis and figure preparation were both performed using R version 4.1.3 with R Studio [44] and the following packages: readxl version 1.4.0 [45], tidyverse version 1.3.1 [46], magrittr version 2.0.3 [47], forcats version 0.5.1 [48], lme4 version 1.1-29 [49], AICcmodavg version 2.3-1 [50], car version 3.1-0 [51], multcomp version 1.4-19 [52], ggplot2 version 3.3.6 [53], plotrix version 3.8-2 [54], ARTool version 0.11.1 [55], and ggpubr [56].

#### 2.10.2. Prolonged Exposure Analysis

Behavioural bioassays were also conducted over a period of 360 min to observe putative neuropeptide-induced long-term changes in cercariae behaviour. Aliquots of 8 µL of 3 mg/mL NP1, NP5, NP7, and NP8 in MilliQ water were each placed in a tube containing 400 µL of water containing cercariae (final concentration of 59 µg/mL) after 100 µL was used for the pre-exposure analysis. MilliQ water functioned as a control. A 100 µL aliquot was taken from each tube and placed onto a glass slide pre-exposure as well as every 90 min post-exposure over 360 min. The videos were recorded for one min and analysed using the methods stated above. There were 9 replicates of each putative neuropeptide and the control. The behavioural data from each putative neuropeptide were compared against the control data across pre- and post-exposure time points. Statistical analyses were conducted as above but with more post-exposure time points in the within-subjects effect. Furthermore, the pH of the neuropeptide solutions at 59 µg/mL was tested using a TPS pH Cube pH-mV-Temp Meter to identify the potential role of pH in inducing behaviour change.

### 2.11. Secondary Structure Analysis

PEP-FOLD3 (http://bioserv.rpbs.univparisdiderot.fr/services/PEP-FOLD3/, accessed on 27 October 2021) was used to predict the secondary structures of all putative neuropeptides tested in the behaviour bioassays. This is an approach to de novo predict the structure of short peptides by assembling the peptide structure using a greedy procedure with Hidden Markov Model-derived structural alphabets [57]. The structure was visualised using VMD 1.9.3 [58].

## 3. Results

### 3.1. Putative Neuropeptide Precursors Identified in Cercariae

A total of 74 proteins were identified from 247 cercarial protein extracts with high confidence MS/MS spectra (Appendix A). These proteins were further annotated by BLASTp, which indicated the presence of five precursors (Sm-npp-1, Sm-npp-5, Sm-npp-14, Sm-npp-17, and Sm-npp-26) of putative neuropeptides that had previously been identified in *S. mansoni* adults [16,59]. From these precursors, three neuropeptides were exclusively predicted by NeuroPred and 10 putative neuropeptides were also confirmed by LC-MS/MS. The three predicted neuropeptides were AFVRL-NH_2_ and GFVRI-NH_2_ (Sm-npp-1) and GLRNMRM-NH_2_ (Sm-npp-14) (Figure 1 and Appendix A). The 10 established putative neuropeptides supported by LC-MS/MS included AAYMDLPW-NH_2_ and AAYIDLPW-NH_2_ (Sm-npp-5), NYLWDTRL-NH_2_ (Sm-npp-17), TLFNPILF-OH, NFDPILF-OH, SYFDPIIY-OH, SYFDPILF-OH, NEDRQFE-OH, and EHFDPIIY-OH (Sm-npp-26), and SMYERIKSIPTE-OH (Sm-npp-14), the last of which was confirmed for the first time by MS/MS. Signal peptides were detected in all precursors except Sm-npp-17.

The mass spectral coverage of 11 further putative neuropeptide precursors, whose discovery in the cercarial state is novel, is displayed in Figure 2 and includes predicted post-translational modifications with high confidence. These precursors were numbered Sm-npp-30-40, following the system of nomenclature used for *S. mansoni* neuropeptide precursor genes. Of these precursors, five (Sm-npp-30, Sm-npp-32, Sm-npp-34, Sm-npp-35, and Sm-npp-36) were previously derived from the *S. mansoni* genome; however, this is the first time they have been confirmed through proteomic analysis in cercariae [17]. Most peptides detected by the LC-MS/MS were consistent with the cleavage sites predicted by NeuroPred. Moreover, five de novo sequenced peptides had similarities to the known neuropeptide family LR/RLamide [20,24], and thus were also listed as putative neuropeptides, which might be translated from open reading frames that have not been included in the current genome version. A total of 30 putative neuropeptides were predicted from these novel precursors (Appendix A). The precursor proteins Smp_122500.1 (Sm-npp-31), Smp_136220.1 (Sm-npp-32), and Smp_147060.1 (Sm-npp-34) did not show characteristics of signal peptides (based on SignalP or PrediSi), suggesting possible incompleteness of the current sequences for these proteins [16].

### 3.2. Protein–Protein Interaction Analysis

In addition to putative neuropeptide precursors, peptidomic analysis identified several proteins with potential catalytic or enzymic function, such as taurocyamine kinase, polyprenol reductase, aldolase, and lysosomal alpha-mannosidase (Appendix A). An additional 13 identified proteins were annotated as uncharacterised proteins, suggesting possible novel functions. A PPI network analysis was performed using all annotated proteins, including all putative neuropeptide precursors except for Sm-npp-39 (Figure 3A). The majority of edges demonstrated an insignificant level of interplay (*p*-value = 0.0577), likely due to the uncharacterised status of almost 40% of identified proteins. Within the predicted interactions, myosin heavy chain (MHC) was determined as the most interactive node as it connected with *DBRE1*, *ALDOA*, caskin-1, taurocyamine kinase, *DNAH1*, immunoglobulin superfamily member 10 (IGSF10), and several other structural proteins.

Of interest, two groups contained three putative neuropeptide precursors predicted to interact with each other based on co-expression values (Appendix A), suggesting possible collaborative functions. These predicted interactions include those between Smp_042120.1 (Sm-npp-1), Smp_150650.1 (Sm-npp-14), and Smp_142160.1 (Sm-npp-17) in addition to those between Smp_004710.1 (Sm-npp-30), Smp_188580.1 (Sm-npp-35), and Smp_143270.1 (Sm-npp-33). Moreover, Smp_004710.1 (Sm-npp-30) was predicted to interact with Smp_121960.1 (Figure 3A), a protein which contains a transmembrane domain and is unique to the Phylum Platyhelminthes. The domains of this protein include several from the immunoglobulin superfamily, and the protein displays strong similarities to *Schistosoma haematobium* IGSF10 and leucine-rich repeat-containing protein 4. Within a PPI network, this protein showed another interaction with an MHC protein. The statistical analysis of the PPI network is shown in Appendix A. Most PPIs had a path length of 3, meaning the nodes were intensely connected, which was also reflected by the majority of the nodes having degrees of 1 or 2. Although the highest betweenness and closeness centrality values were 1.00, most neighbours had relatively low centrality values.

The gene expression levels corresponding to the identified putative neuropeptide precursor proteins were derived from a previous study which compared the transcriptomes of different stages of *S. mansoni* including cercariae, immature adults, and schistosomule stages 1, 2, and 3 [36]. The relative expression of protein nodes and precursors of putative neuropeptides are compared in Figure 3B. Most of the genes had higher levels of expression in both cercariae and immature adults, suggesting relatively high relevance to these two stages (Appendix A).

### 3.3. Analysis of Putative Neuropeptides on Schistosoma mansoni Cercaria Behaviour

A total of nine putative neuropeptides, derived from precursor proteins with high conservation among *Schistosoma* and high expression in the cercarial stage, were selected to be tested for bioactivity by external application to cercariae (Table 1). The novel putative neuropeptide precursors identified had negligible homology with species outside of the *Schistosoma* genus and a closely related digenetic trematode, *Trichobilharzia regenti* (Appendix A). Additionally, all putative neuropeptide precursors were relatively highly expressed in the cercarial and/or immature schistosome stage (Figure 3B). Finally, smaller putative neuropeptides (<2 kilodaltons) were prioritised due to the predicted faster rate of penetrating cercariae. Selected putative neuropeptides included four from known neuropeptide precursors: NP1 (Sm-npp-5); NP2 (Sm-npp-26); NP4 (Sm-npp-14); and NP8 (Sm-npp-17). This list also included five putative neuropeptides from novel neuropeptide precursors: NP3 (Sm-npp-38); NP5 (Sm-npp-33); NP6 (Sm-npp-36); NP7 (Sm-npp-30); and NP9 (Sm-npp-35). Although meeting our criteria, NP2 was ultimately not tested because its excessive hydrophobicity prevented it from dissolving in MilliQ water.

Acute exposure analyses were performed at a concentration of 3 mg/mL for all putative neuropeptides (Appendix A). Of the eight putative neuropeptides tested, NP1, NP5, NP7, and NP8 all exclusively affected active-state behaviour, including increasing brief stopping and spinning on the spot followed by directional changes. Additionally, body-first movement, which is uncommon in the absence of chemical stimuli, also became more frequent post-exposure [60] (Appendix A). These changes were quantitatively supported by significant decreases in active velocity and significant increases in active angular SD. Average active velocity significantly decreased by 24.37% post-exposure to NP5 (*p*-value = 0.0011), 9.08% post-exposure to NP7 (*p*-value = 0.0025), and 20.64% post-exposure to NP8 (*p*-value *=* 0.0315) (Table 2 and Figure 4A). Average active angular SD significantly increased by 36.46% post-exposure to NP1 (*p*-value = 0.0120), 37.57% post-exposure to NP5 (*p*-value = 0.0009), and 77.92% post-exposure to NP8 (*p*-value < 0.0001) (Figure 4B). Overall, all four novel putative neuropeptides were implicated, to at least some extent, in a behavioural shift with spinning and directional changes resulting in greater wanderingness and lower effective velocities when active.

An acute exposure analysis was also conducted with NP1, NP5, NP7, and NP8 at 0.1 mg/mL to determine cercarial sensitivity to the putative neuropeptides (Appendix A). NP7 and NP8 did not produce any significant change in behaviour at this concentration (Appendix A). NP1 produced a significant 126.69% increase in the average relative number of passive points (*p*-value *=* 0.0034), whereas it did not induce a significant change in this metric at 3 mg/mL (*p*-value *=* 0.2046). NP5 produced comparable increases in spinning, stopping, and body-first movement at 0.1 mg/mL to at 3 mg/mL. This was denoted by a 42.57% significant increase in average active angular SD (*p*-value *=* 0.0034) (Appendix A). This suggests higher cercarial sensitivity to NP5 than to the other putative neuropeptides. It is unclear why NP1 induced different behaviour changes at 0.1 mg/mL.

The effects of the NP1, NP5, NP7, and NP8 were additionally tested in a prolonged exposure analysis over 360 min incubated at 59 µg/mL, with periodic observations every 90 min (Appendix A). There were negligible differences in pH between the neuropeptide solutions, suggesting that pH was unlikely to significantly affect cercarial behaviour change (Appendix A). Only NP7 failed to produce significant changes in any behavioural metrics over 360 min (Table 3). The inability of NP7 to induce significant behaviour changes at 0.1 mg/mL or over 360 min suggests lower cercarial sensitivity to it than to NP1, NP5, or NP8. In contrast to the acute exposure experiment, NP1 and NP8 did not cause consistent significant decreases in active-state velocity or increases in active angular SD; instead, both caused a significant increase in passive behaviour. NP1 exposure produced significant increases in average relative passive points, increasing by 122.96% at 90 min post-exposure (*p*-value *=* 0.0038) and 67.44% at 270 min post-exposure (*p*-value *=* 0.0144) (Figure 5). This data is consistent with the acute exposure experiment at 0.1 mg/mL, however not at 3 mg/mL. NP8 exposure also significantly increased passive behaviour, increasing the average relative number of passive points by 225.05% at 90 min (*p*-value *=* 0.0001), 194.58% at 180 min (*p*-value *=* 0.0031) and 107.84% at 270 min (*p*-value *=* 0.0020) post-exposure. These increases were likely due to significant increases in average relative passive states, which increased by 236.20% at 90 min (*p*-value *=* 0.0004) and 141.77% at 180 min (*p*-value *=* 0.0076). This indicates that the increases in passive behaviour following NP1 and NP8 exposure were most substantial within the first 90 min post-exposure and gradually diminished thereafter. These observations suggest that exposure to NP1 and NP8 may accelerate cercarial activity to shift from ‘active’ to ‘passive’ with prolonged exposure, while prolonged exposure to NP7 does not cause any significant change.

In contrast to NP1 and NP8, prolonged exposure to NP5 produced similar effects to acute exposure, including consistently affected active behaviour, significantly decreasing average active velocity, and significantly increasing average active angular SD at all time points post-exposure. NP5 exposure caused significant decreases in average active velocity of 21.57% at 90 min (*p*-value *=* 0.0055), 35.68% at 270 min (*p*-value *=* 0.0065), and 43.94% at 360 min (*p*-value *=* 0.0014) post-exposure. Average active angular SD significantly increased by 126.38% at 90 min (*p*-value *=* 0.0008), 167.93% at 180 min (*p*-value *<* 0.0001), 232.55% at 270 min (*p*-value *<* 0.0001), and 231.66% at 360 min (*p*-value *<* 0.0001) post-exposure. This is consistent with the behavioural changes induced in the acute experiments at 3 mg/mL and 0.1 mg/mL. Qualitative observations indicate that prolonged exposure increases “head-pulling” behaviour, in which the cercariae is pulled forward by its extended head while tail movement only shifts its direction (Appendix A). The findings of the acute and prolonged exposure experiments suggest that NP5 exposure primarily affects active behaviour without significantly influencing passive behaviour. Qualitatively, no putative neuropeptides were observed to induce increased cercarial tail shedding during the prolonged exposure assay.

### 3.4. Structural Analysis of Bioactive Putative Neuropeptides That Modify Schistosoma mansoni Cercaria Behaviour

Further peptidomic analysis was performed on NP1, NP5, and NP8 due to their bioactivity. NP1 was perfectly conserved across several *Schistosoma* spp., suggesting functional specificity within the genus (Figure 6A). The peptide also displayed a high tendency towards turn helices and high hydrophobicity. NP5 was also highly hydrophobic and well-conserved across human-infective schistosomes (*S. mansoni*, *Schistosoma japonicum* and *S. haematobium*) (Figure 6B). Secondary structure analysis was performed on NP5, demonstrating a high propensity of α-helices throughout the whole sequence. NP8 was neutral, also highly hydrophobic, and well-conserved across *Schistosoma* spp. (Figure 6C). Descriptions of the other putative neuropeptides used in the acute exposure experiments are presented in Appendix A.

## 4. Discussion

This study aimed to further elucidate putative neuropeptides in *S. mansoni* cercariae and investigate their potential to regulate cercarial behaviour. This was achieved through peptidomic analyses of cercarial peptide extracts in conjunction with behavioural bioassays. The results indicated that four novel putative neuropeptides stimulated significant change in cercarial behaviour and thus may be considered biocontrol target candidates.

The *S. mansoni* genome enabled the prediction of 19 putative neuropeptides [16], yet little is known about their functions. In this study, we expanded the list of putative neuropeptides by 30, derived from 11 precursor proteins whose discovery in cercariae was novel (see Appendix A). All novel precursors were supported by high confidence MS/MS spectra (containing at least five consecutive *b* or *y* ions). A comparative analysis using BLASTp identified negligible homologues outside of the Phylum Platyhelminthes (see Appendix A). For example, Sm-npp-36 did not have any homolog outside of the class Trematoda, while Sm-npp-33 was exclusive to *Schistosoma* (*E*-value < 0.0001). This suggests that these putative neuropeptides could be used as targets for the manipulation of cercarial physiology and behaviour without negatively impacting non-target sympatric species outside of the class Trematoda. Genes encoding the identified putative neuropeptide precursors were also relatively highly expressed in cercariae and immature adults [36], implying that the derived putative neuropeptides might play roles in biological processes specific to these two stages.

All short peptides derived from putative neuropeptide precursors possessed strong hydrophobic regions containing amino acids well conserved in *Schistosoma* and across other trematode species. Hydrophobic interactions are believed to play crucial roles in the interplay between putative neuropeptides and receptors [61,62]. This suggests that target receptors for these putative neuropeptides probably have relatively high hydrophobic binding capacity. This is supported by observations of putative neuropeptides—of comparable size and hydrophobicity to those employed in this study—traversing the membrane to enter the cells of parasite *Trypanosoma brucei* and causing death [63]. Moreover, it is possible that the peptides identified in this study could aggregate to form macrostructures via hydrophobic interactions, as has been reported for fibrillated neuropeptides that can regulate secretion pathways [64].

Acute and prolonged exposure assays were conducted to observe the effect of eight putative neuropeptides on cercarial behaviour. NP1 and NP8, derived respectively from Sm-npp-5 (Smp_052880.1) and Sm-npp-17 (Smp_142160.1), produced similar behavioural changes to each other in both acute and prolonged exposure analyses. During acute exposure at 3 mg/mL, these changes included increased spinning, stopping, and body-first movements, which contrast with the more commonly observed tail-first movement along consistent trajectories (Appendix A). In contrast, prolonged exposure to these two putative neuropeptides over a period of 360 min produced an increase in the frequency of passive-state behaviour without affecting active state behaviour, relative to the control. This suggests that these putative neuropeptides may accelerate a transition to increasingly passive behaviour; however, it is unclear why acute and prolonged exposure produce such differing behaviour changes. Both putative neuropeptides had strong structural similarities, including turn-helical structures and hydrophobic regions, which suggest a higher probability of being able to penetrate the tegument and bind to cercarial internal receptors.

Putative neuropeptides NP5 and NP7, derived from the Sm-npp-33 (Smp.143270.1) and Sm-npp-30 (Smp_004710.1) precursor proteins, respectively, produced similar increases in spinning, stopping, and body-first movement to NP1 and NP8 during acute exposure at 3 mg/mL. However, NP7 did not cause significant behaviour changes at 0.1 mg/mL or during the prolonged exposure analysis, suggesting lower cercarial sensitivity to its exposure. In contrast to the other putative neuropeptides, NP5 induced similar changes at both 3 mg/mL and 0.1 mg/mL, suggesting higher cercarial sensitivity (see Appendix A). Furthermore, during prolonged exposure analysis, body-first movement and spinning became increasingly frequent over 360 min. NP5 uniquely produced prolonged effects exclusively on cercariae active-state behaviour. It has been speculated that active- and passive-state behaviours are controlled by different mechanisms; thus, it is unsurprising that exposure to a particular putative neuropeptide could primarily change behaviour in one specific state [41]. Furthermore, prolonged-exposure cercariae also displayed “head-pulling” behaviour which has previously been observed in cercariae when proximate to mammalian skin [65]. This suggests that NP5 may be involved in the host-seeking and penetration process.

Because of the lack of predicted interactions between the putative neuropeptide precursors and other proteins, it is difficult to speculate on the mechanisms through which the putative neuropeptides affect cercarial behaviour. The first of two clusters of predicted neuropeptide precursor protein interactions are between Sm-npp-1, Sm-npp-17, and Sm-npp-14 (the latter two of which respectively contain NP4 and NP8). It may be speculated that the inability of NP4 to penetrate the epithelium was due to its lower relative hydrophobicity. In contrast, the second cluster, which contained Sm-npp-30, Sm-npp-33, and Sm-npp-35 (respectively containing NP7, NP5, and NP9), was predicted to interact with an MHC via the transmembrane protein IGSF10, which contains domains associated with regulating neuron migration and synaptic assembly, organisation, and adhesion [66,67]. MHC in *S. mansoni* adult muscle has been observed to contribute significantly to striated muscle-like thick filaments that combine with smooth muscle-like thin filaments to form a smooth muscle architecture [68]. The MHC gene is highly expressed in cercariae, suggesting functional specificity to this stage, and has been considered a vaccine candidate [69]. MHC is also predicted to interact with several other structural protein families present as nodes with high degrees, such as titin [70,71] and collagen (types I and V) [72], which have been well characterised in schistosomes. Because almost all putative neuropeptide precursor proteins employed in this study contain a signal peptide, it may be speculated that the interaction between these precursors and muscle-related proteins occurs extracellularly. The predicted interactions between the cluster of putative neuropeptide precursors and these muscle-relevant proteins, via two or more pathway steps, implies that the derived neuropeptides could play roles in regulating muscle function.

Future experimentation with the putative neuropeptides identified in this study is recommended to confirm them as *S. mansoni* cercariae neuropeptides and elucidate their underlying mechanism of activity. This may be achieved by gene-specific mRNA in situ hybridisation and antibody-mediated immunolocalisation, with particular focus on spatial localisation within neuronal cells. Moreover, the identification and spatial expression of transmembrane-bound cognate receptors could be investigated. This study used an exogenous application of putative neuropeptides; therefore, we expect that those peptides stimulating responses may interact with receptors on the surface of cercarial sensory epithelia cells. Schistosomes contain numerous receptors, including GPCRs and ionotropic receptors, that are yet to be deorphanized [73]. Another method to investigate the behavioural role of these putative neuropeptides would be to downregulate expression of their encoding genes, as was performed successfully in a study on the soybean parasitic nematode *Heterodera glycines* [74].

## 5. Conclusions

In conclusion, this study revealed putative neuropeptides derived from 16 precursor proteins, including 11 precursor proteins discovered for the first time in cercariae, through peptidomic analysis of *S. mansoni* cercariae. Through behavioural bioassays, we showed that acute exposure to AAYMDLPW-NH_2_, NRKIDQSFYSYY-NH_2_, FLLALPSP-OH, and NYLWDTRL-NH_2_ caused increased spinning, stopping, and body-first movement in the cercariae active state. Prolonged exposure to AAYMDLPW-NH_2_ and NYLWDTRL-NH_2_ increased passive-state behaviour, while prolonged exposure to NRKIDQSFYSYY-NH_2_ caused progressively increased spinning, directional change, and head-pulling behaviour in cercariae, similar to that observed when proximate to mammalian skin. These putative neuropeptides were specific to the Phylum Platyhelminths, with several others being exclusive to *Schistosoma*, making them potential targets for biocontrols to induce accelerated energy depletion or disorientation in cercariae. These results may inform future neurobiological innovations for inhibiting *S. mansoni* cercariae infectivity and thereby mitigating the spread of schistosomiasis.

## Figures and Tables

**Figure 1 biology-11-01344-f001:**
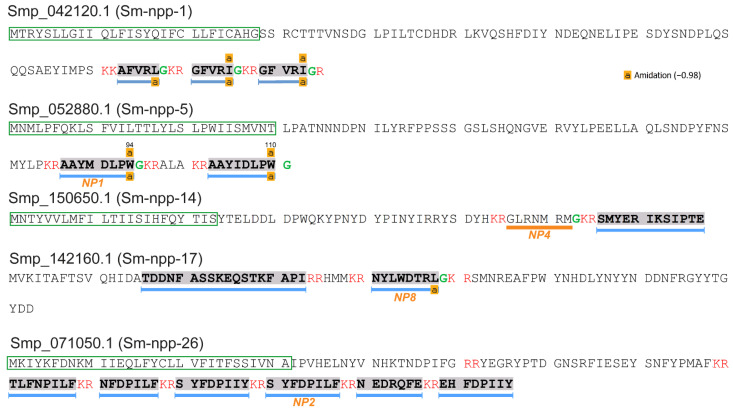
Identification of neuropeptide precursors previously reported in adult *S. mansoni* [16,59]. The peptide segments, supported by high-confidence MS/MS spectra (supported by at least 5 consecutive *b* or *y* ions), are displayed in grey shade and underlined with blue bars. Predicted signal peptides are shown in a green frame; cleavage sites are in red font; glycine residues available for C-terminal amidation are in green font. Putative neuropeptides employed in behavioural bioassays are underlined in orange with their corresponding descriptions.

**Figure 2 biology-11-01344-f002:**
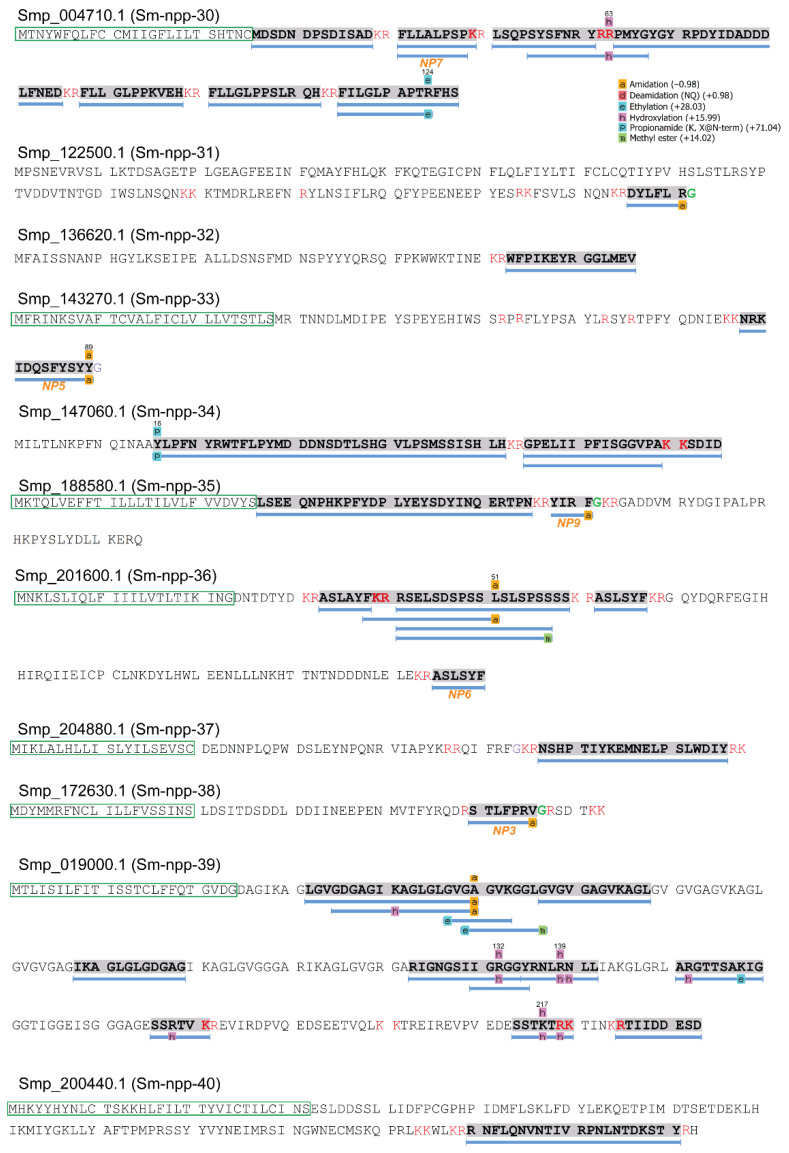
The MS/MS spectral coverage of novel putative neuropeptide precursors in *S. mansoni* cercariae. The peptide segments, supported by high-confidence MS/MS spectra (supported by at least 5 consecutive *b* or *y* ions), are displayed in grey shade and underlined with blue bars. The PTMs were predicted by the PEAKS studio based on the database search results. Predicted signal peptides are shown with a green frame; cleavage sites are in red font; glycine residues available for C-terminal amidation are in green font. Putative neuropeptides employed in the behavioural bioassay are underlined with their corresponding descriptions.

**Figure 3 biology-11-01344-f003:**
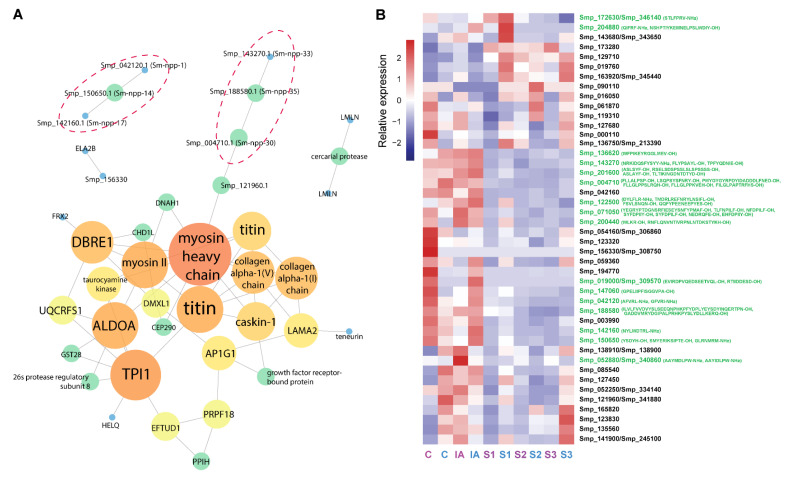
PPI and gene expression analysis of identified *S. mansoni* cercarial proteins. (**A**) PPI analysis (confidence score below 0.15) with putative neuropeptide precursors circled in red. For colour coding: nodes are on a spectrum from high degree (red) to low degree (blue). The size of the circles also corresponds with node degree. (**B**) The relative gene expression of all identified proteins, including precursors of putative neuropeptides (values are transcripts per million units, TPMs) at different stages: cercariae (C), immature adult (IA), and schistosomule stages 1, 2, and 3 (S1, S2, and S3). This was constructed from a previous report, hence some proteins were not included [36]. The corresponding gene IDs of the *S. mansoni* genome v7 are shown after “/”. The sequences of predicted putative neuropeptides are shown in parentheses. Colour codes of labels: purple, female; blue, male; green, precursors of putative neuropeptides. Cercariae of individual sex were extracted from *B. glabrata* snails with monomiracidial infections [36]. Data on the expression of these genes in miracidia and sporocysts is present in Appendix A. This gene expression data could not be directly compared due to the differences in experiments used to identify the data.

**Figure 4 biology-11-01344-f004:**
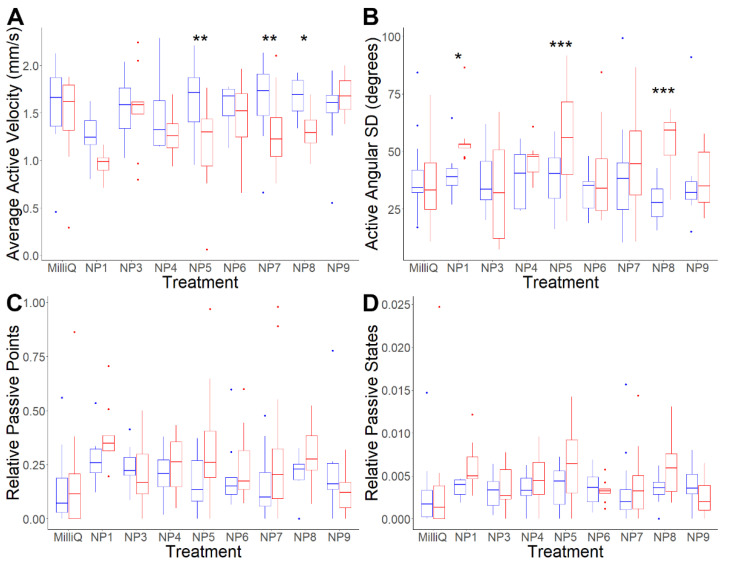
Changes in *S. mansoni* cercariae behaviour from one min pre- and post-exposure to MilliQ water and 3 mg/mL NP1, NP3, NP4, NP5, NP6, NP7, NP8 and NP9. (**A**) Average active velocity (mm/s); (**B**) active angular SD (degrees); (**C**) relative passive points; (**D**) relative passive states. Boxplot indicates median, 25th and 75th percentiles, minimum and maximum data with outliers represented by dots. A two-way ANOVA test was used to calculate *p*-values for the mixed-effects interaction of pre- and post-exposure to putative neuropeptide treatments against MilliQ water: * *p*-value *<* 0.05, ** *p*-value < 0.01, *** *p*-value < 0.001. Colour: blue: pre-exposure; red: post-exposure.

**Figure 5 biology-11-01344-f005:**
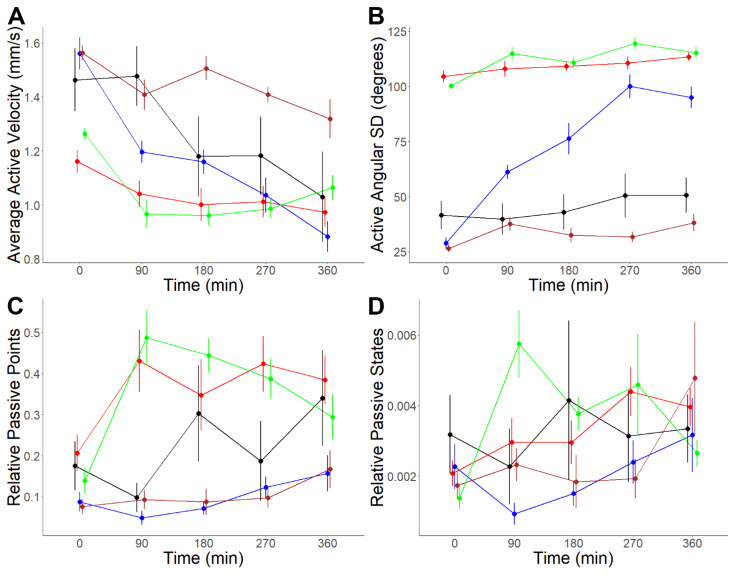
Changes in *S. mansoni* cercariae behaviour pre-exposure and 90 min, 180 min, 270 min, and 360 min post-exposure to MilliQ water and 3 mg/mL NP1, NP5, NP7 and NP8 (final concentration: 59 µg/mL). (**A**) Average active velocity (mm/s); (**B**) active angular SD (degrees); (**C**) relative passive points; (**D**) relative passive states. A two-way ANOVA test was used to calculate *p*-values for the mixed-effects interaction of pre-exposure and at different durations post-exposure to putative neuropeptide treatments against MilliQ water. Statistical analysis and figure configuration were both performed using R Studio. Colour: black: MilliQ; red: NP1; blue: NP5; brown: NP7; green: NP8.

**Figure 6 biology-11-01344-f006:**
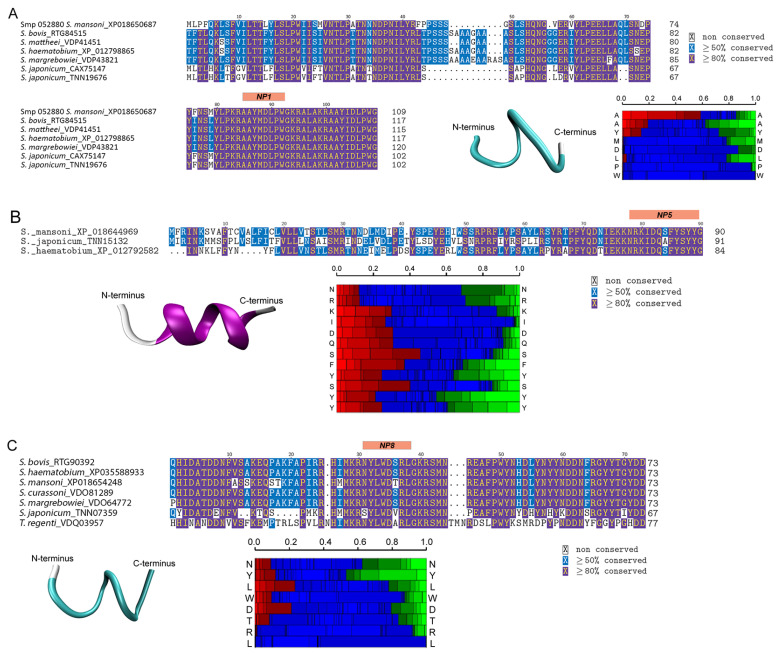
Structure of bioactive peptides and comparative multiple sequence analyses of their respective precursor proteins, including information on conserved amino acids. (**A**) Sm-npp-5 and NP1 (AAYMDLPW-NH_2_); (**B**) Sm-npp-33 and NP5 (NRKIDQSFYSYY-NH_2_); and (**C**) Sm-npp-17 and NP8 (NYLWDTRL-NH_2_). Structural alphabet (SA) (vertical axis) at each position. Colour code: red: helical, green: extended, blue: coil. Heatmaps and structural models were generated using PEP-FOLD3 and visualised using VMD 1.9.3.

**Table 1 biology-11-01344-t001:** Summary of putative neuropeptides synthesised.

ID	Sequence	Molecular Weight (Da)	Accession of Precursor
NP1	AAYMDLPW-NH_2_	965.131	Smp_052880.1 (Sm-npp-5)
NP2	SYFDPILF-OH	1001.137	Smp_071050.1 (Sm-npp-26)
NP3	STLFPRV-NH_2_	817.979	Smp_172630.1 (Sm-npp-38)
NP4	GLRNMRM-NH_2_	876.110	Smp_150650.1 (Sm-npp-14)
NP5	NRKIDQSFYSYY-NH_2_	1582.721	Smp_143270.1 (Sm-npp-33)
NP6	ASLSYF-OH	686.756	Smp_201600.1 (Sm-npp-36)
NP7	FLLALPSP-OH	857.052	Smp_004710.1 (Sm-npp-30)
NP8	NYLWDTRL-NH_2_	1079.214	Smp_142160.1 (Sm-npp-17)
NP9	YIRF-NH_2_	596.724	Smp_188580.1 (Sm-npp-35)

**Table 2 biology-11-01344-t002:** Two-way ART ANOVA results, comparing the effects of *S. mansoni* cercarial exposure to NP1, NP3, NP4, NP5, NP6, NP7, NP8, and NP9 at 3 mg/mL in MilliQ water one min pre- and post-exposure. Test statistics, degrees of freedom, and *p*-values for the main effects of putative neuropeptide and period (pre- vs. post-exposure), and their interaction. If the interaction effect was significant, pairwise contrasts are included; testing analysed the significance of the change in behaviour pre- vs. post-exposure between MilliQ and each of the putative neuropeptides. Metrics: active velocity, active angular SD, relative number of passive points and states. Significant data (*p*-value < 0.05) is in bold.

Metrics	Effect	Statistic	df	*p*-Value	MQ: NP1	MQ: NP3	MQ: NP4	MQ: NP5	MQ: NP6	MQ: NP7	MQ: NP8	MQ: NP9
Active velocity	Neuropeptide	2.98	8	**0.0049**								
	Period	53.69	1	**<0.0001**								
	Neuropeptide:Period	4.79	8	**<0.0001**	0.0793	0.4597	0.6763	**0.0011**	0.9736	**0.0025**	**0.0315**	0.1182
Active angular SD	Neuropeptide	2.05	8	**0.0480**								
	Period	22.35	1	**<0.0001**								
	Neuropeptide:Period	3.67	8	**0.0009**	**0.0120**	0.9753	0.1447	**0.0009**	0.3849	0.0959	**<0.0001**	0.2574
Relative passive points	Neuropeptide	2.91	8	**0.0058**								
	Period	12.50	1	**0.0006**								
	Neuropeptide:Period	1.55	8	0.1496	-	-	-	-	-	-	-	-
Relative passive states	Neuropeptide	3.33	8	**0.0020**								
	Period	7.13	1	**0.0088**								
	Neuropeptide:Period	1.21	8	0.3017	-	-	-	-	-	-	-	-

**Table 3 biology-11-01344-t003:** Two-way ART ANOVA results, comparing the effects of *S. mansoni* cercarial exposure to NP1, NP5, NP7, and NP8 at 3 mg/mL to MilliQ water pre-exposure and at 90 min intervals over 360 min post-exposure (final concentration: 59 µg/mL). Test statistics, degrees of freedom, and *p*-values for the main effects of putative neuropeptide and period (pre-exposure vs. 90 min, 180 min, 270 min, and 360 min post-exposure), and their interaction. If the interaction effect was significant, pairwise contrasts testing analysed the significance of the change in behaviour pre- vs. post-exposure between MilliQ and each of the putative neuropeptides. Metrics: active velocity, active angular SD, relative number of passive points and states. Significant data (*p*-value < 0.05) is in bold.

Metrics	Effect	Statistic	df	*p*-Value	Interaction	0–90	0–180	0–270	0–360
Active velocity	Neuropeptide	30.64	4	**<0.0001**					
	Period	21.84	4	**<0.0001**					
	Neuropeptide:Period	2.93	16	**0.0003**	**NP1:MilliQ**	0.5295	0.7231	0.5637	0.3432
					**NP5:MilliQ**	**0.0055**	0.0694	**0.0065**	**0.0014**
					**NP7:MilliQ**	0.2462	0.1611	0.4193	0.7179
					**NP8:MilliQ**	**0.0198**	0.2809	0.5734	0.4158
Active angular SD	Neuropeptide	181.59	4	**<0.0001**					
	Period	29.43	4	**<0.0001**					
	Neuropeptide:Period	7.67	16	**<0.0001**	**NP1:MilliQ**	0.6446	0.8506	0.3824	0.4818
					**NP5:MilliQ**	**0.0008**	**<0.0001**	**<0.0001**	**<0.0001**
					**NP7:MilliQ**	0.1627	0.9820	0.2415	0.6917
					**NP8:MilliQ**	**0.0384**	0.4700	0.3075	0.9527
Relative passive points	Neuropeptide	20.60	4	**<0.0001**					
	Period	6.47	4	**<0.0001**					
	Neuropeptide:Period	3.00	16	**0.0002**	**NP1:MilliQ**	**0.0038**	0.3417	**0.0144**	0.3564
					**NP5:MilliQ**	0.8690	0.2362	0.7278	0.4844
					**NP7:MilliQ**	0.5104	0.3731	0.9642	0.6251
					**NP8:MilliQ**	**0.0001**	**0.0031**	**0.0020**	0.5326
Relative passive states	Neuropeptide	3.68	4	**0.0121**					
	Period	2.44	4	**0.0492**					
	Neuropeptide:Period	2.06	16	**0.0127**	**NP1:MilliQ**	0.2428	0.2149	0.1153	0.4699
					**NP5:MilliQ**	0.6458	0.9770	0.9498	0.6564
					**NP7:MilliQ**	0.2899	0.8596	0.8572	0.6859
					**NP8:MilliQ**	**0.0004**	**0.0076**	0.0702	0.7113

## Data Availability

Data are contained within the article or Appendix A.

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
