# Peer review of "Identification of Putative Neuropeptides That Alter the Behaviour of Schistosoma mansoni Cercariae"

_biology, 2022, doi:10.3390/biology11091344_

Round 1

Reviewer 1 Report

The manuscript describes the identification of putative neuropeptides altering the behavior of Schistosoma mansoni cercariae. The manuscript is well-written and the data are informative in this field but the reviewer found several points needed to be corrected and clarified before the decision.

 [Major concern]

1. The authors applied synthetic neuropeptides externally and observed the behavior of the cercariae. Do the authors consider autocrine and/or paracrine of the peptides? Is there any evidence showing that these neuropeptides are secreted from cercariae to swimming water?

 2. The authors need to discuss about the pH of neuropeptide solutions.

 3. Did the neuropeptides have any effects on tail shedding of cercariae in prolonged exposure analysis?

 4. Line 263: 8 ml of 3 mg/ml neuropeptide in 508 ml solution will not be 59 mg/ml. The authors need to correct the description.

 5. Figure 3 (line 345-346): How did the authors distinguish female and male?

 6. Table 2: What are the numbers in bold? If the numbers < 0.05 are shown in bold, there are several other numbers need to be in bold in the table.

 7. Table 3: It was difficult to comprehend the table for the reviewer. Why there is no effect, statistics, df and p-value for NP8? Why the df of NP7 is “16”?

[Minor concern]

1. “S. mansoni” in figure legends needs to be in italic.

 2. Figure 4: The results of Pre-exposure (Grey) are too light to see.

 3. Figure 5: The colors of lines are difficult to see, especially red (orange?) and amber (yellow?). It would be better to insert the explanation in the figure.

 4. Revise line 257.

 5. The authors analyzed neuropeptides in miracidia previously. Is it possible to include the data in figure 3B even it would be difficult to distinguish female and male miracidia?

Author Response

Thank you very much for reviewing our manuscript, we have made revisions to the manuscript following the reviewer's comments. Please see below our point-by-point response to the queries raised by the reviewer. 

  1. The authors applied synthetic neuropeptides externally and observed the behavior of the cercariae. Do the authors consider autocrine and/or paracrine of the peptides? Is there any evidence showing that these neuropeptides are secreted from cercariae to swimming water?

RESPONSE: These putative neuropeptides have not, to our knowledge, been identified as secreted outside of the cercariae. We applied the neuropeptide solutions externally because it was not feasible to inject them into the cercariae. We are currently unsure which cells are affected by the putative neuropeptides or in which cells they are produced. Therefore, we could not identify whether the behaviour changes were due to autocrine or paracrine signalling.

  1. The authors need to discuss about the pH of neuropeptide solutions.

RESPONSE: The pH of the different neuropeptide solutions at their final concentrations in the behavioural bioassays (59 µg/mL) were all calculated as between 7.00 and 7.60, suggesting that it is unlikely that pH is a significant contributing factor to the induced behaviour changes. The differences in pH are now acknowledged in the Methods and Results (Lines 296-297 and 466-467). We lacked access to equipment which could determine the pH of the stock solutions at 3 mg/mL; however, we do not think it is necessary to identify the pH at this concentration.

  1. Did the neuropeptides have any effects on tail shedding of cercariae in prolonged exposure analysis?

RESPONSE: No consistent tail shedding was observed in acute or prolonged exposure analyses. This detail has been added to the results (Line 514-516)

  1. Line 263: 8 ml of 3 mg/ml neuropeptide in 508 ml solution will not be 59 mg/ml. The authors need to correct the description.

RESPONSE: Thank you for pointing out our lack of clarity. An 8 µL aliquot of peptide solution was added after 100 µL (of the initial 500 µL) was used in the pre-exposure sample, so the new volume after adding the peptide was 408 µL. Adding 8 µL of 3 mg/mL neuropeptide solution to 400 µL would therefore mean dividing the concentration by 51, so the final concentration would be 59 µg/mL. I have added this detail to the methods (Line 286-289)

  1. Figure 3 (line 345-346): How did the authors distinguish female and male?

RESPONSE: Cercariae of individual sex were extracted from B. glabrata snails with monomiracidial infections. I have added this detail to the Figure legend. (Line 381-382)

  1. Table 2: What are the numbers in bold? If the numbers < 0.05 are shown in bold, there are several other numbers need to be in bold in the table.

RESPONSE: Thank you for noting this oversight. This detail has been added to Table 2 and Table 3 legends and all P < 0.05 data has been bolded. (Line 452 and 502)

  1. Table 3: It was difficult to comprehend the table for the reviewer. Why there is no effect, statistics, df and p-value for NP8? Why the df of NP7 is “16”?

RESPONSE: I have added in a specification that the final five columns refer to the interaction effects between Milli-Q water and the neuropeptides between the different timepoints (Line 502)

[Minor concern]

  1. “S. mansoni” in figure legends needs to be in italic

RESPONSE: This has been fixed in all figure legends.

  1. Figure 4: The results of Pre-exposure (Grey) are too light to see.

RESPONSE: The colours have been changed to blue and red for clearer viewing and contrast (Line 438).

  1. Figure 5: The colors of lines are difficult to see, especially red (orange?) and amber (yellow?). It would be better to insert the explanation in the figure.

RESPONSE: Clearer colours with improved contrast have been implemented (Line 486).

  1. Revise line 257.

RESPONSE: This revision has been implemented (Line 281-282).

  1. The authors analyzed neuropeptides in miracidia previously. Is it possible to include the data in figure 3B even it would be difficult to distinguish female and male miracidia?

RESPONSE: Gene expression data on miracidia and sporocysts is now displayed in a representative supplementary figure (Line 632-637) and as raw quantitative data in Table S1 (Line 685-687). Because this data was attained from a different experiment, it could not accurately be presented in Figure 3 alongside cercariae and intramammalian schistosome data. Unlike with cercariae and schistosomula, we were not able to identify sex-specific data on gene expression in miracidia and sporocysts.

Reviewer 2 Report

This study identifies putative neuropeptides of Schistoma mansoni, and assesses behavioural alternation in cercariae exposed to these peptides. Identification of behavioural-altering neuropeptides can innovate the potential biocontrols of the infective stage of this serious zoonotic fluke to prevent human schistosomiasis in future. The manuscript is well organized and written well, and worthy of publication of the journal. A limited number of minor revision is necessary before publication as shown below (all are editorial corrections):

1.     L112: What is “acidic acid”?

2.     L188: “(<2 kDa)” (?)

3.     L204: Superscript “®” in “StarFrost®”

4.     L257: “… version 0.11.1 [55] and ggpuber [56].”

5.     L283-284, L317, L337, L344, L406: “S. mansoni” in italic.

6.     L294: “precursors”, and “in” in roman.

7.     L411-412, L416, Table 2, L424-446, L464, L472-476: “P” in italic in “P-value”

8.     L421-440: Anterior to this page, citations of Figure and Table are shown by bold font, but from this page this style becomes changed. Please keep consistency. Similarly, in other pages, for example, L480-494.

9.     L486-492: Scientific names at five sites in italic.

10.  L513: “Platyhelminthes” in roman.

11.  L515: “(E-value <0.0001)” <<< “E” in italic. Similarly at L645.

12.  L518, L612: “Trematoda” in roman.

13.  L684-847: Variable styles of description are seen, for example abbreviated journal names vs. fully spelled journal names, scientific names of organisms in roman vs. in italic.

14.  Please thoroughly check the manuscript by the authors.

Author Response

Thank you very much for reviewing our manuscript, we have made revisions to the manuscript following the reviewer's comments. Please see below our point-by-point response to the queries raised by the reviewer. 

  1. L112: What is “acidic acid”?

RESPONSE: Thank you for the correction. It should be “glacial acetic acid” (Line 124).

  1. L188: “(<2 kDa)” (?)

RESPONSE: this has been changed to “kilodalton” for clarity (Line 204 & 417).

  1. L204: Superscript “®” in “StarFrost®”

RESPONSE: Fixed (Line 230).

  1. L257: “… version 0.11.1 [55] and ggpuber [56].”

RESPONSE: Fixed (Line 282).

  1. L283-284, L317, L337, L344, L406: “ mansoni” in italic.

RESPONSE: Fixed throughout manuscript

  1. L294: “precursors”, and “in” in roman.

RESPONSE: Fixed (Line 323)

  1. L411-412, L416, Table 2, L424-446, L464, L472-476: “P” in italic in “P-value”

RESPONSE: Fixed throughout manuscript

  1. L421-440: Anterior to this page, citations of Figure and Table are shown by bold font, but from this page this style becomes changed. Please keep consistency. Similarly, in other pages, for example, L480-494.

RESPONSE: Fixed throughout manuscript

  1. L486-492: Scientific names at five sites in italic.

RESPONSE: Fixed (Line 521-526)

  1. L513: “Platyhelminthes” in roman.

RESPONSE: Fixed (Line 545)

  1. L515: “(E-value <0.0001)” <<< “E” in italic. Similarly, at L645.

RESPONSE: Fixed (Line 193, 547 and 688)

  1. L518, L612: “Trematoda” in roman. Fixed.

RESPONSE: Fixed (Line 546, 549 & 626)

  1. L684-847: Variable styles of description are seen, for example abbreviated journal names vs. fully spelled journal names, scientific names of organisms in roman vs. in italic.

RESPONSE: Formatting issues highlighted above have all been addressed.

  1. Please thoroughly check the manuscript by the authors.

RESPONSE: Authors have checked the manuscript and approved.

Reviewer 3 Report

I would focus on the behavior part of the manuscript and have a few comments.

1. The scrambled peptide negative control is missing. It is even more important as the authors are using relatively high concentrations of the neuropeptides.

2. Please specify how pixel coordinates were calculated into the distance in mm. 

3. For the angular SD measurements. It is not clear to me if (how) you extracted (selected) straight and wandering tracks. Or did you measure the turning angles between the points of the track? Please specify.

4. The representative videos are showing mostly the single cercaria sporadically appearing in the field of view. My point is that the animals are not tracked a significant amount of the time because they are not in the field of view. 

It might be beneficial to limit the animal to a certain arena where individuals can be tracked full time. In this case, we eliminate the problem of measuring the different individuals without noticing it.

5. Maybe more supplementary videos (two-three per condition) are needed to get a full impression of the effect of the neuropeptide.

6.  For the figures (Figures 4 and 5) describing the quantifications of behavior three different graphs depict the passive state (points) or passive state duration are excessive. Please eliminate one at least.

Author Response

Thank you very much for reviewing our manuscript, we have made revisions to the manuscript following the reviewer's comments. Please see below our point-by-point response to the queries raised by the reviewer. 

  1. The scrambled peptide negative control is missing. It is even more important as the authors are using relatively high concentrations of the neuropeptides.

RESPONSE: A scrambled peptide negative control was not used in this experiment. However, peptides which induced significant changes in acute exposure at 3 mg/mL (final conc when added to the assay: 37-99 µM) were also tested at 0.1 mg/mL (final conc. 1-2 µM) to determine sensitivity. The use of a scrambled peptide is not something we have used previously for semiochemical assays, nor other questions relating to peptide functional analysis, yet we now recognise the importance of such a negative control and will implement in future assays.

  1. Please specify how pixel coordinates were calculated into the distance in mm. 

RESPONSE: With the magnification the bioassays were conducted at, 87 pixels corresponds to 1 mm, therefore 1 pixel is roughly equal to 11.5 µm. The dimensions of the FOV have been added (Line 233-234). Scale bars have been added to all supplementary videos for perspective.

  1. For the angular SD measurements. It is not clear to me if (how) you extracted (selected) straight and wandering tracks. Or did you measure the turning angles between the points of the track? Please specify.

RESPONSE: Angular SD is derived from the standard deviation of the change in angles between consecutive points in a track, indicating the magnitude and frequency of directional change. I have added this detail to the Methods (Line 266-269)

  1. The representative videos are showing mostly the single cercaria sporadically appearing in the field of view. My point is that the animals are not tracked a significant amount of the time because they are not in the field of view. It might be beneficial to limit the animal to a certain arena where individuals can be tracked full time. In this case, we eliminate the problem of measuring the different individuals without noticing it.

RESPONSE: The cercariae need to be on hydrophilic slides to prevent the water from becoming deep, which causes the cercariae to become unclear even within the FOV and producing unreliable results. On these slides, even 10 µL of water would expand outside the FOV. With our set-up, we are unable to have both enough water for sufficient cercariae to analyse and a low enough quantity of water in which to view them all in the FOV without the water being too deep and making the cercariae difficult to track.

  1. Maybe more supplementary videos (two-three per condition) are needed to get a full impression of the effect of the neuropeptide.

RESPONSE: We have added two-three supplementary videos per neuropeptide treatment (Line 655-684). The only exception is NP7, for which it was difficult to find multiple supplementary videos showing its behaviour changes.

  1. For the figures (Figures 4 and 5) describing the quantifications of behavior three different graphs depict the passive state (points) or passive state duration are excessive. Please eliminate one at least.

RESPONSE: Average duration of passive states has been removed from Figures 4 and 5 (lines 438 & 486), as well as Tables 2 and 3 (lines 452 & 502), since it is not necessary alongside the relative passive points metric.

Round 2

Reviewer 3 Report

Thank you. All questions are addressed.